# scRNA-seq and scATAC-seq analyses highlight the role of TNF signaling pathway in chronic obstructive pulmonary disease model mice

Qiang Zhang[1‡], Li Zhou[2,3‡], Lindong Yuan[4‡], Ruihua Zhang[2,5‡], Shanshan Pan[2], Xizi Wang[1], Lili Yi[1], Fengjiao Yuan[1], Xianchao Guo[6,7], Mingliang Gu[1], Yan Wang[4*], Xiaodong Jia[1,8*]

1 Joint Laboratory for Translational Medicine Research, Liaocheng People's Hospital, Liaocheng, Shandong, China, 2 BGI-Qingdao, BGI-Shenzhen, Qingdao, China, 3 Beijing Easyresearch Technology Limited, Beijing, China, 4 Department of Respiratory and Critical Care Medicine, Liaocheng People's Hospital, Liaocheng, Shandong, China, 5 College of Life Sciences, University of Chinese Academy of Sciences, Beijing, China, 6 Harbin Inji Technology Co., Ltd., Harbin, Heilongjiang, China, 7 GenVista Technology Co., Ltd, Harbin, Heilongjiang, China, 8 School of Life Science and Technology, Shandong Second Medical University, Weifang, Shandong, P.R.China

‡ These authors contributed equally to this work.
* jiaxiaodong1018@163.com (XJ); wangyan6178@163.com (YW)

## Abstract

Chronic obstructive pulmonary disease (COPD) is a prevalent and progressive form of respiratory disease in which patients exhibit persistent respiratory damage affecting the alveoli and/or airway due to exposure to toxic gases or particulate matter. C57BL/6 mice were exposed to cigarette smoke and lipopolysaccharide to establish a COPD model mice, followed by scATAC (Assay for Transposase Accessible Chromatin) sequencing and scRNA sequencing of lung tissue samples. The resultant data revealed consistent findings between scATAC-seq and scRNA-seq regarding cell types, differentially expressed genes, and signaling pathways in COPD model mice. Tumor necrosis factor (TNF) signaling pathway enrichment was evident in the scRNA-seq and scATAC-seq datasets, with similar trends in monocytes/macrophages, dendritic cells, and B cells. In COPD model mice, significant tumor necrosis factor receptor 1 (TNFR1) upregulation and high levels of activity related to cellular communication were observed, and significant increases in *Il1b*, *Csf1*, and *Bcl3* site accessibility were evident in cells. These findings suggest that the TNF signaling pathway maybe associated with COPD.

## Introduction

Chronic obstructive pulmonary disease (COPD) is a prevalent, progressive respiratory disease characterized by airway and alveolar abnormalities and restricted airflow due to exposure to toxic gases or particulate matter [1,2]. Cigarette smoke, chronic air pollution, and bacterial infections are the three primary risk factors for COPD, with

**Data availability statement:** The data that support the findings of this study have been deposited into CNGBdb (China National GeneBank DataBase). The link to data set: https://db.cngb.org/search/project/CNP0004399/

**Funding:** This study was supported by The National Natural Science Foundation of China (Grant No. 82000066), and the Project of Medical and Health Science and Technology in Shandong Province(Grant No.202303021557) and the Traditional Chinese Medicine Science and Technology Project of Shandong Province (Grant No.Z-2023033). The funders had no role in study design, data collection and analysis, decision to publish, or preparation of the manuscript.

**Competing interests:** The authors have declared that no competing interests exist.

the most extensive research focusing on cigarette smoke [3,4]. Cigarette smoke exposure induces oxidative stress [5–6], activating lung resident cells like epithelial cells and alveolar macrophages. This activation triggers the release of chemokines and cytokines, including pro-inflammatory mediator TNF-α, recruiting lymphocytes and neutrophils to the airways [7,8]. COPD subtypes, such as small airway disease, chronic bronchitis, and emphysema, vary in symptoms and treatment responses. Emphysema, notably, involves destructive enlargement of air spaces in the distal terminal bronchioles [9–12].

COPD onset and progression involve various cellular processes such as autophagy, apoptosis, mitochondrial and metabolic dysfunction, senescence, extracellular matrix proteolysis, DNA damage, and immune cell infiltration [13–15]. Single-cell RNA sequencing identified a COPD-specific alveolar type II epithelial cell subpopulation with unique HHIP (Hedgehog-interacting protein) expression and aberrant stress tolerance profiles. Endothelial cells show overlapping and distinct transcriptional profile shifts, possibly contributing to vascular inflammation and injury susceptibility in COPD [16]. As the most disease-, aging-, and smoking-relevant cell types, monocytes, club cells, and macrophages exhibit changes leading to alveolar epithelium dysfunction [17]. The populations with more differentially expressed genes were monocytes, macrophages, and ciliated epithelial cells in the COPD lungs [18]. In COPD patients, alveolar macrophages exhibit transcriptional plasticity, characterized by increased levels of invading and proliferating cells, reduced cellular motility, altered lipid metabolism, and mitochondrial dysfunction [19]. The gene sets upregulated in COPD samples were related to the neutrophilic inflammatory response and TNF-α activation of the NF-κB signaling pathway [20]. TNF is a key regulator of inflammation in the cytokine network. TNF-α, a proinflammatory cytokine, is associated with disease progression, and it plays a vital role in regulating various inflammatory pathways involving NF-κB [21–23]. Blocking TNF-α signaling, decreased breast cancer metastasis to lungs in mice by ∼60%, and decreasing TNF-α signaling, decreased NF-κB activation and the expression of inflammation-related genes that regulate metastasis [24]. Tumor necrosis factor receptor 2 converts the tumor inhibiting ability of TNF-α into a tumor advocating factor, thereby directly promoting the proliferation of some types of cancers such as lung, breast, and colon cancer [25].

The establishment of appropriate animal models can accurately simulate the pathological characteristics of human COPD, which is conducive to the development of effective intervention and treatment in the short term. By analyzing the experimental results of COPD mice models, we hope to conduct comparative studies with human diseases, so that we can more effectively understand the occurrence and development of COPD and discover new therapeutic targets. Here, smoke exposure and lipopolysaccharide (LPS) treatment were used to establish a murine model of COPD. Lung tissue samples were then collected from these animals and used to conduct single-cell ATAC sequencing (scATAC-seq) and single-cell RNA sequencing (scRNA-seq) analyses to elucidate the mechanistic roles of specific cell types in COPD pathogenesis. These experiments revealed related changes in lung structural

cell of murine model of COPD, and along with a concomitant increase in immune cell levels and the enrichment of corresponding inflammation-, apoptosis-, and oxidative stress-related signaling pathways. TNF signaling pathway enrichment was evident in both the scATAC-seq and scRNA-seq datasets, with comparable trends in mononuclear cells/macrophages, B cells, and dendritic cells (DCs).

## Materials and methods

### Animal model establishment

This study included 4 C57BL/6 COPD model mice (2 male, 2 female; 8 weeks old) and 2 C57BL/6 control mice (1 male, 1 female; 8 weeks old). Control mice: on day 1 and day 15, each mouse was anesthetized, intubated with a tracheal tube, and infused with 30 μl of normal saline; on days 2–14 and 16–29, mice were maintained under normal conditions. COPD model mice: on day 1 and day 15, each mouse was anesthetized, intubated with a tracheal tube, and infused with 25 μg of LPS in 30 μl of normal saline (the methods were adapted from Chen *et al*.) [26], no smoking on the day of the LPS infusion; On days 2–14 and days 16–29, mice underwent continuous cigarette smoke fumigation: 4 cigarettes per session, with each session lasting 1 h, conducted once daily (the methods were adapted from Duan *et al*. and Cavarra *et al*.) [27,28]. In each fumigation session, mice inhaled smoke from four large cigarettes, Shanghai Tobacco Group Co. Ltd. (Tar 10 mg, nicotine 0.8 mg, smoke carbon monoxide 12 mg). Mice were anesthetized using intraperitoneal injections of Pentobarbital sodium (50 mg/kg). The thoracic cavity was opened for lung tissue harvesting, then sutured, and mice were euthanized through cervical dislocation. A portion of the lung tissue was fixed in 4% neutral formaldehyde for 48 h, embedded in paraffin, sectioned, and stained with hematoxylin and eosin (HE). The bronchial injury in the lung and the infiltration of inflammatory cells in the bronchial and lung tissues were observed under the microscope (Fig 1a). The remaining lung tissues underwent scRNA-seq and scATAC-seq analysis. These mouse were purchased from East China Normal University, and housed in environmentally controlled pathogen-free conditions throughout the experiments. All protocols follow the animal care guidelines of East China Norma University (Laboratory animal production permit: SCXK-2016–0004; Laboratory animal use permit: SXYK-2020–0015). According to the Declaration of Helsinki, our study was approved by the Ethics Committee of Liaocheng People's Hospital (No. 2020021).

### Single-cell suspension preparation

Lung tissues were sectioned into approximately 0.5-mm³ pieces in RPMI-1640 medium (GIBCO, C22400500BT) supplemented with 1% Penicillin/Streptomycin (GIBCO, 15140–122) and enzymatically digested using the MACS

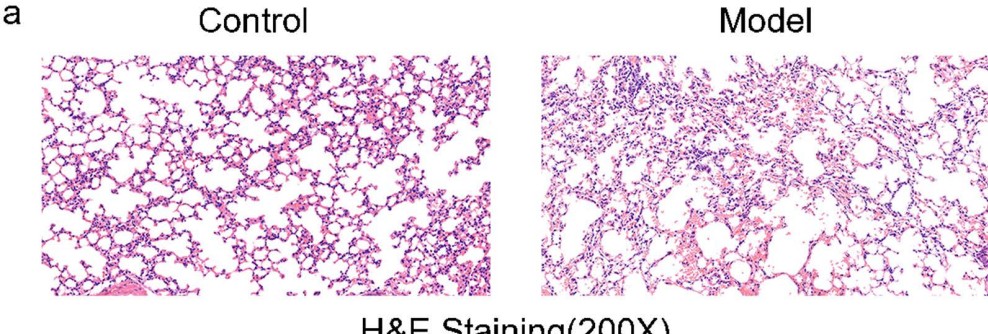

**Fig 1. Pathological examination of the mouse lung tissues.** The mouse lung tissues were fixed, sectioned at 4 μm thickness, and stained with H&E solution, magnification ×200. The alveolar dilatation was obvious, and some of the alveolar septal rupture showed blistering emphysema. There was a small amount of inflammatory cell infiltration near the small bronchial branches (terminal bronchioles, etc.), mainly lymphocytoplasmic cell infiltration.

Tumor Dissociation Kit_mouse (Miltenyi Biotec, 130-096-730) at 37°C for 30 min with agitation, following the manufacturer's guidelines. Dissociated cells were filtered through 70-µm and 40-µm cell strainers (BD, 352340) and centrifuged at 300 g for 10 min. The supernatant was discarded, and the cell pellet was resuspended in red blood cell lysis buffer (Thermo Fisher), followed by a 2-min incubation on ice to lyse red blood cells. After washing twice with PBS (THERMO, 10010023), the cell pellets were resuspended in PBS (containing 0.04% BSA), readying them for single-cell RNA sequencing. The cell requirements: Cell viability: ≥ 90%, Clumping rate: ≤ 5%, and the background without many impurities.

## scRNA-seq with DNBelab C4 system

The DNBelab C Series Single-Cell Library Prep Set (MGI) was used for single-cell RNA-seq library preparation, as described previously [29]. Briefly, single-cell suspensions were processed into barcoded scRNA-seq libraries *via* droplet encapsulation, emulsion breakage, mRNA-captured bead collection, reverse transcription, and cDNA amplification and purification. The cDNA product had a concentration > 20 ng/µL with a volume of 30 µL, and a peak for the fragment size distribution ranges between 800 bp to 2000 bp. Indexed sequencing libraries were constructed according to the manufacturer's protocol. The sequencing libraries were quantified by Qubit ssDNA Assay Kit (Thermo Fisher Scientific). Libraries were sequenced on the MGI 2000 sequencer. The sequencing employed a paired-end read structure: Read 1 comprised 30 bases, including two 10-bp cell barcodes and a 10-bp unique molecular identifier (UMI), while Read 2 captured 100 bases of the transcript sequence.

## Nucleus suspension preparation

Cell pellets were resuspended in 100 µL homogenization buffer containing 20 mM Tris pH 8.0 (AMBION, AM9855G), 250 mM sucrose (BBI, A610498-0500), 0.1% IGEPAL® CA-630 (SIGMA, I8896-50ML), 0.2 U/µL RNase inhibitor (MGI), 1 × protease inhibitor cocktail (ROCHE, 4693116001), 1% bovine serum albumin (BBI, A600332-0100), and 0.1 mM DTT (SIGMA, 646563) and incubated on ice for 3 min to lyse the cell membrane. Nuclei were pelleted by centrifugation at 500 × g for 5 min at 4°C, then resuspended in blocking buffer containing 1% BSA and 0.2 U/µL RNase inhibitor in 1 × PBS for Single-Cell ATAC-seq preparation. The nuclear requirements were: Nuclear integrity rate of ≥ 90%, Clumping rate of ≤ 10%, Nucleus yield of > 100,000, and the background without many impurities.

## scATAC-seq with DNBelab C4 system

The DNBelab C Series Single-Cell ATAC Library Prep Set (MGI, 1000021878) was used for scATAC-seq library preparation. Lung tissue cell suspensions were prepared, and nuclei were extracted following previously established protocols [30]. Then, ~ 100,000 nuclei and 25 µL of a transposition reaction mixture containing 10 mM TAPS-NaOH (pH 8.5), 5 mM MgCl2, 10% DMF, and 4 µL of in-house Tn5 transposes were combined for Tn5 tagmentation. The resultant transposed single-nuclei suspensions were then used to prepare barcoded scATAC-seq libraries *via* droplet encapsulation, pre-amplification, emulsion breakage, captured bead collection, DNA amplification, and purification. The DNA product had a concentration > 10 ng/µL with a volume of 100 µL, and a peak for the fragment size distribution ranges between 200 bp to 800 bp. Indexed libraries were prepared according to the manufacturer's protocol. Concentrations were measured with a Qubit ssDNA Assay Kit. Libraries were sequenced on a BGISEQ-500 sequencer with the following sequencing strategy: 50-bp read length for Read 1 and 76-bp read length for Read 2.

## scATAC-seq data pre-processing

Raw fastq files were filtered and trimmed using PISA (v 0.0.0.9999) (https://github.com/shiquan/PISA) with settings (-q 20 -dropN, -mode Tn5 -p) and aligned to the mm10 using BWA (v 0.7.17) [31]. Reads were filtered to eliminate individual

unmapped reads, paired unmapped reads, multi-unmapped reads, mitochondrial reads, and duplicates using bap2 (v 0.6.6) (https://github.com/caleblareau/bap) (-r mm10 -bt CB --mapq 30). This approach yielded 166,533,130 read fragments from 6 murine lung samples, averaging 27,755,521 reads per sample.

### scATAC-seq quality control

Quality control (QC) and downstream analyses of scATAC-seq data were performed with a previously published pipeline [32] and the ArchR toolkit (v 1.0.2) [33]. Initially, samples were subjected to individual processing and annotation to explain any source-specific differences in quality. The ArchR object of each sample was generated by the createArrowFiles() function with TileMatParams = 500. Cells were filtered based on TSS enrichment (filterTSS = 4), unique fragments (filterFrags = 1000), and doublets (using filterDoublets with filterRatio = 1) (S1 Table).

### scATAC-seq batch alignment and dimensionality reduction

Following QC, an ArchR projection of all samples was created using ArchRProject(), with the genome version set to mm10. Dimensionality reduction was conducted using addIterativeLSI(), focusing on the top 30 dimensions. Batch effect correction was applied using the addHarmony() function with default parameters. Clusters were identified at a resolution of 0.8 and visualized using uniform manifold approximation and projection (UMAP) embedding techniques.

### scATAC-seq peak annotation and motif identification

Following ArchR projections, a tile matrix was generated, mapping cells and a 500 bp genomic window to columns and rows, respectively. Confident peaks for each cell cluster were called using addReproduciblePeakSet() and annotated with the annotatePeak function from ChIPseeker [34] with default settings. Then, the addMotifAnnotations function was used to identify motifs with reference to the CIS-BP Database [35].

### scATAC-seq differential gene, peaks, and transcription factor (TF) analyses

Subpopulations within cell compartments and sample sets were identified by assessing differentially accessible genes, peaks, and TFs using the appropriate ArchR quantification matrix (GeneScoreMatrix, PeakMatrix). This was performed with the ArchR getMarkerFeatures function *via* the "Wilcoxon" method.

### Peaks co-accessibility analyses

The Cicero tool [36] within ArchR assessed peak co-accessibility using default settings for addCoAccessibility() and getCoAccessibility(), with results visualized *via* plotBrowserTrack (upstream = 100,000, downstream = 100,000).

### Transcription factor footprinting

TF footprints were used for the characterization of TF occupancy using composite functions. Initially, relevant motif positions were obtained with getPositions(), after which motifs of interest were obtained with getFootprints(), and plotting was performed using plotFootprints with the default parameters.

### scRNA-seq raw data processing

Following read alignment to the mm10 reference genome, Cell Ranger was used to compute gene counts with default parameters. Doublets were removed using DoubletFinder [37], which computes the average transcriptional profiles of pairs of randomly chosen cells to generate pseudo-doublets and then identifies candidate doublets based on the similarity of a given cell to these pseudo-doublet pairs.

### scRNA-seq cell clustering and identification

Cells were pre-processed to retain those expressing at least 500 genes observed in a minimum of three cells and to remove cells with over 10% mitochondrial DNA content before performing downstream analyses. Global clustering of the lung tissue dataset was conducted using the Seurat package (v4.0.3) for R (v4.0.2) [38]. The NormalizeData function was used to normalize data across replicates using the default settings, followed by FindVariableFeatures and the vst method to identify the 2,000 genes exhibiting the greatest variability. Variables genes exhibiting consistency in different replicates were selected for batch correction with the FindIntegrationAnchors function followed by an integrated analysis. Clustering and visualization were conducted using Seurat (https://satijalab.org/seurat/articles/integration_introduction.html) with default parameters based on the results of integrated analyses.

### DEG identification and analysis

DEGs across cell types within the tissue were identified and analyzed using the FindMarkers and FindAllMarkers functions in Seurat. The FindAllMarkers function was used to analyze DEGS in different types of cells in a given tissue, and DEGs were identified based on established criteria ($P<0.05$, FC>2). We used the Wilcoxon rank-sum test, the default setting in Seurat, for differential expression analysis using the FindAllMarkers function. The enrichGO/enrichKEGG function in clusterProfiler [39] was used for Gene Ontology (GO)/KEGG enrichment analyses.

### Cell-cell communication analyses

Cell-cell communication analysis was performed using the CellPhoneDB program (v 1.1.0) [40] (www.cellphonedb.org) based on the scRNA-seq dataset. Further analyses were performed only for ligands and receptors expressed by a minimum of 10% of a given cell type, with interactions being designated as not evident in cases where either ligands or receptors were not detectable. Average expression levels for ligand-receptor pairs were compared among cell types, and those exhibiting a $P<0.05$ were retained for predicting cell-cell communication within groups.

### scATAC-seq and scRNA-seq dataset integration

ScATAC-seq and scRNA-seq datasets were integrated with the addGeneIntegrationMatrix() function, the addGeneIntegrationMatrix function from the ArchR package (version 1.0.2) was used for gene integration analysis. Using gene expression values and accessibility scores for cells meeting quality criteria.

## Results

### scATAC-seq-based characterization of changes in chromatin accessibility of COPD model mice

A comprehensive approach was employed herein to characterize the mechanisms governing the pathogenesis of COPD in murine lungs (C57BL/6 mice). Initially, the chromatin accessibility landscape was profiled at the single-cell level *via* scATAC-seq, as in prior reports [41,42] (Fig 2a). To mitigate gender- or individual-specific differences, an equal gender ratio was maintained during testing [43]. Lung tissues from four COPD model mice and two control mice were processed to obtain single-cell suspensions, from which nuclei were extracted and treated with Tn5 transposase. The quality of scATAC-seq data was assured by evaluating transcriptional start site (TSS) enrichment, mitochondrial DNA contamination, fragment size distribution, and doublet exclusion. In total, 22,038 nuclei that passed the quality filter were identified, corresponding to 12,158 cells from the COPD model mice and 9,880 from the control mice. These yielded a median TSS enrichment score of up to 26, a median of 4610 fragments, and 251,024 peaks that were called, with >50% of the fragment fraction overlapping with these peaks. These data exhibited appropriate fragment size periodicity, with >30% of fragments located at TSS and a high degree of correlation between the aggregate profiles within each group. The estimated doublet/multiple percentages in each group were 3.8% on average, and the corresponding data were removed

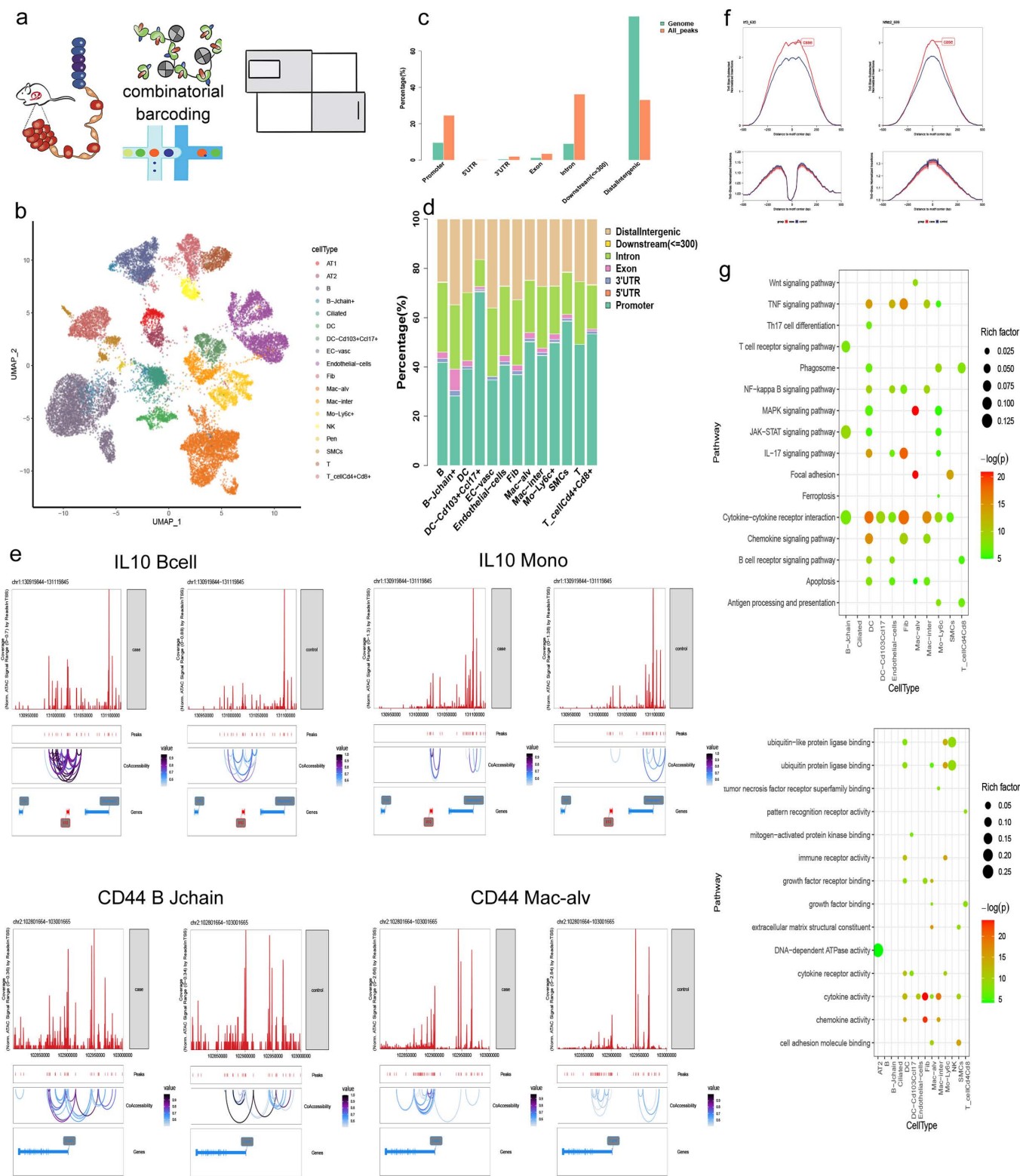

**Fig 2. scATAC-seq data analysis of the mouse lung tissues.** a: The two approaches used for this project are scATAC-seq (top) and scRNA-seq (bottom). b: UMAP plot showing the integrated cell profiles of scATAC-seq from both case and control mouse lung. The dots indicate individual cells, and the

cell-type identity is indicated by color. c: Distribution comparison of mouse genome features and calling peaks annotations. Genome features including Promoter, 5′ UTR, 3′ UTR, Exon, Intron, Downstream and the DistalIntergenic; d: Distribution of peaks annotation on the mouse genome of each cell type. Genome features including Promoter, 5'UTR, 3'UTR, Exon, Intron, Downstream, and DistalIntergenic. Colors represent different cell types. e: The track visualization of genome-wide *cis*-regulatory interaction networks for certain regions across specific genes for specific cell types between case and control groups. The gene and cell type are indicated on the head of each figure. The bin size of each window was located in the top left corner. The are color represents different peak accessibility degrees. f: The motif footprints signal of specific TF within 4 kb windows. Red represents the case group, while blue represents the control group. g: The Bubble Chart showing KEGG enrichments for markers of each cell type in the case (upper) and control (down) groups. The color represents the *P*-value, and cell types are indicated on the x-axis.

from further analysis. Collectively, these results indicated consistent enrichment across open chromatin regions, providing a comprehensive dataset for further analysis.

To explore molecular and cellular heterogeneity related to COPD development more effectively, subsequent analyses were performed with the ArchR pipeline [44]. Dimensionality reduction was initially performed with TF-IDF, identifying 31 cell clusters at a resolution of 0.8. Distance-weighted accessibility models were then utilized to calculate a gene score matrix, converting the degree of chromatin accessibility into a corresponding measure of gene expression. Established clusters were then identified based on cell type-specific marker patterns using marker collections from sources including the CellMarker [45], Mouse Cell Atlas (MCA) [46], and PanglaoDB [47] databases as well as published analyses of murine tissues, particularly studies focused on murine lings. Ultimately, these 31 clusters were classified into 4 cell groups, including epithelial, endothelial, immune, and stromal cells, with 18 cell subtypes (Fig 2b), in line with published profiles from mouse lungs [48,49]. AT1 and AT2 cells, alveolar macrophages, and J-chain+ B cells were identified based on *Ager, Sftpb, Mcemp1*, and *Jchain* expression, respectively (S1 Fig). Following these cell type determinations, peaks with significant difference across cell types were calculated and compared with the distribution of total features on the genome, revealing a marked increase in the accessibility of promoter and intronic regions in the lung tissues (Fig 2c). Analysis of peak regions and associated genes across 18 cell types (S2 Table) identified 1333 differential peaks in B cells, 6563 in endothelial cells, and 6188 in alveolar macrophages, with significant differences assessed by Wilcoxon test (false discovery rate [FDR] < 0.05, |log2FC| > 0.5). Notably, the proportion of peaks in promoter regions in CD103 + dendritic cells accounts for 70%. And in most immune cells, the distribution of peak numbers in the promoter region exceeds 40%, showing a predominance in immune cells(Fig 2d, S2 Fig). This suggests a high degree of transcriptional activity associated with these regions, prompting further comparisons of total peaks with the genome to clarify trends in COPD-related accessibility changes in COPD model mice.

A cis-regulatory interaction network, constructed using data from immune cell subsets, revealed consistent peak patterns across different cell types, with stronger signals in the COPD model mice (Fig 2e, S3 and S4 Figs). Notably, IL-10, an immunomodulatory cytokine with anti-inflammatory properties linked to various disease processes [50], showed increased fragment coverage and complex co-accessibility in the COPD mice, especially in monocytes and B cells. In contrast, CD44 co-accessibility spanned a larger region and was relatively dispersed, although it exhibited similar between-group variance. Differential signals were also evident between COPD model mice and control mice in the B cell and alveolar macrophage populations. Given that TFs (transcription factors) are essential regulators of diverse cellular processes, motif identification was performed based on the differentially accessible peaks in these different groups, revealing multiple TFs that exhibited significant differences (Fig 2f). These TFs showed higher binding signals in the COPD model mice, consistent with higher levels of chromosome accessibility and the activation of associated regulatory mechanisms.

Comparative analysis of COPD model mice and control mice across all samples identified 28,139 genes with significant differences based on the gene score matrix (GeneScoreMatrix), with 17,965 upregulated and 10,174 downregulated, predominantly enriched in immune-associated and universally essential pathways (Fig 2g). Both the TNF and

cytokine-cytokine receptor interaction pathways were enriched in most cell types, consistent with their broadly important roles in shaping diverse processes, including proliferation, apoptosis, differentiation, and immunoregulatory activity in response to corresponding activating stimuli and receptor-ligand binding events. Including the Th17 cell differentiation and B cell receptor signaling pathways, were enriched in immune cell populations. Enriched GO biological process terms for these DEGs included protein binding, cytokine activity, and energy supplementation. Collectively, these findings from scATAC-seq analyses indicate enhanced cellular and intercellular activity in lung cells from the COPD model mice.

**scRNA-seq-based the pulmonary immune-related transcriptomic landscape of COPD model mice**

Lungs from four COPD model mouse and two control mouse were harvested and digested for scRNA-seq analyses. Twenty-one cell types were identified based on molecular markers, including CD4 T cells, CD8 T cells, general T cells, B cells, J-chain$^+$ B cells, proliferating lymphocytes, NK cells, monocytes, interstitial macrophages, alveolar macrophages, CD209$^+$ DCs, CD103$^+$/CCL17$^+$ DCs, ciliated cells, Col13a1$^+$ fibroblasts, Col14a1$^+$ fibroblasts, smooth muscle cells, vascular endothelial cells, Vcam1$^+$ endothelial cells, Pen cells, AT1 cells, and AT2 cells (Fig 3a, S5 Fig). Relative to the control mice, the COPD model mice showed marked reductions in the relative abundance of AT2 cells, vascular endothelial cells, Vcam1$^+$ endothelial cells, ciliated cells, and pen cells. In contrast, there were significant increases in the levels of B cells, monocytes, and NK cells, together with slight increases in CD4 T cell, CD8 T cell, and J-chain$^+$ B cell abundance (Fig 3b). Analyses of the proportions of differential genes expressed in these different cell types revealed many DEGs in B cells, T cells, monocytes, alveolar macrophages, and NK cells, the majority of which were upregulated (Fig 3c). This was consistent with the overall cell type trends detected in this experiment. AT2, Col13a1$^+$ fibroblast, and Col14a1$^+$ fibroblast cells only showed upregulated DEGs, while ciliated cells, smooth muscle cells, and Vcam1$^+$ endothelial cells exhibited fewer upregulated DEGs, and vascular endothelial cells exhibited more upregulated DEGs (Fig 3c).

Enrichment analyses revealed the enrichment of DCs for neutrophil degranulation, regulation of cytokine production, and cellular responses to external stimuli signaling pathways (Fig 3d). Monocytes were enriched for the positive regulation of cell death, neutrophil degranulation, inflammatory response, regulation of cytokine production, and cellular responses to external stimuli signal pathways (Fig 3d). Interstitial and alveolar macrophages were enriched to regulate cytokine production, inflammatory response, and cellular responses to external stimuli pathways (Fig 3d). CD4$^+$ T cells were enriched for the positive regulation of cell death, regulation of cellular response to stress, and apoptosis signaling pathways (Fig 3d). CD8$^+$ T cells were enriched for the leukocyte differentiation, neutrophil degranulation, regulation of cellular response to stress, and cellular responses to external stimuli signaling pathways (Fig 3d). B cells were enriched for the positive regulation of cell death, neutrophil degranulation, phagosome, and cellular responses to external stimuli signal pathways (Fig 3d). Endothelial cells were enriched for the positive regulation of cell death and cellular responses to external stimuli signaling pathways (Fig 3d). Col13a1$^+$ and Col14a1$^+$ fibroblasts were enriched for the cellular response to stress and reactive oxygen species metabolic rate, Oxidative Stress, Redox, and cellular responses to external stimuli signaling pathways (Fig 3d). Ciliated cells were only enriched for the cellular responses to external stimuli pathway (Fig 3d).

Further differential TF expression analysis indicated downregulation of *Sox7*, *Hoxa5*, *Klf*, and *Foxf1*, linked to tracheal, alveolar, and lung mesenchymal development and differentiation in the COPD model mice. In contrast, increases in the expression of *Foxp1*, *Ebf*, and *Pou2af1* associated with B cells and activated DCs were evident in COPD model mice (Fig 3e). This suggests that alveolar and epithelial/endothelial cell development and differentiation are disrupted in COPD, whereas DCs and B cells are activated in this pathological setting. This may contribute to the induction of adaptive immunity, lymphatic follicular hyperplasia, and persistent lung tissue inflammation, resulting in progressive alveolar destruction and increasingly limited airflow.

Cell interaction analyses revealed that the increased interaction strengths between T cells, DCs, and other cell types in COPD model mice, suggesting that both smoke and LPS exposure may stimulate DC activation (Fig 3f, S6 Fig). This activation may lead to the immune and inflammatory cell response, further implicating successive activation of AT2 cells and

**Fig 3. scRNA-seq data analysis of the mouse lung tissues.** a: UMAP plot showing scRNA-seq profiles of cells in the lung. The dots indicate individual cells, and the cell-type identity is indicated by color; b: Bar plot showing the cell type composition in each sample of the scRNA-seq (top). UMAP plot

showing scRNA-seq profiles of cells. The dots indicate individual cells, and the group identity is indicated by color (bottom); c: Distribution of percentage of cell in each cell type for DEG between case and control; d: The heatmap showing GO enrichments for each cell type DEG between case and control; e: The heatmap showing TF in case and control; f: Distribution of percentage of Cell interaction in each cell type.

fibroblasts. These activated cells trigger the activation of damage-related molecular patterns associated with inflammatory responses, contributing to the release of assorted cytokines, chemokines, acute phase proteins, and antimicrobial peptides. In COPD model mice, the enhanced communication between AT2 cells, fibroblasts, and other cell types suggests that immune cell-regulated inflammatory signaling induced renewed AT2 and fibroblast differentiation and associated alveolar regeneration (Fig 3f, S6 Fig).

## Integrated analyses of scATAC-seq and scRNA-seq datasets of COPD model mice

Integrated analyses of scATAC-seq and scRNA-seq datasets form COPD model mice provided a comprehensive view of transcriptional processes in matched lung tissue samples, confirming uniform data integration without batch effects using ArchR (Fig 4a). Cell type classification was then performed using the integrated data (Fig 4b), identifying 21 cell types based on the expression of specific genes (S3 Fig). Genetic analyses for these 21 cell types revealed highly consistent trends with respect to the changes evident in the scRNA-seq and scATAC-seq datasets in COPD model mice (Fig 4c). Color-based visualization of all samples in these two datasets confirmed excellent concordance between the scRNA-seq and scATAC-seq classification, with only relatively limited individual differences. Differential gene analysis and KEGG pathway enrichment indicated that the TNF signaling pathway might be associated with COPD incidence in COPD model mice (Fig 4d). TNFR1 upregulation in alveolar macrophages and active cellular communication was observed, along with scATAC-seq data showing increased accessibility of *Bcl3*, IL1b, *Fos*, and *Csf1* loci, aligning with scRNA-seq findings of increased expression in COPD model mice compared to control mice(Fig 4e).

## Discussion

This study underscores significant transcriptional changes, along with changes in chromatin dynamics and the epigenetic state of cells, as demonstrated by scRNA-seq and scATAC-seq analyses in COPD model mice. There are consistent findings of cell types, differentially expressed genes, and signaling pathways with COPD model mice by scRNA-seq and scATAC-seq analyses (Fig 2, Fig 3). The TNF signaling pathway was obviously enriched in the scRNA-seq and scATAC-seq datasets (Fig 4d).

The elastase and cigarette smoke exposure animal models have informed our knowledge about the major mechanistic paradigms leading to COPD: inflammation, oxidative stress, protease/antiprotease balance, alveolar cell apoptosis, early senescence, and autophagy [51]. Compared to the control mice, there were increased lung immune cell proportions, and corresponding pathway enrichment in the inflammatory response, cell stress, apoptosis, and phagocytosis in COPD model mice by scRNA-seq analyses (Fig 3d). These were reflected by the high levels of promoter accessibility in several immune cell populations in our COPD model mice by scATAC-seq analyses, compared to the control mice (Fig 2d, Fig 2g). Cigarette smoking is a major cause of COPD, and inhalation of cigarette smoke causes inflammation of the airways, airway wall remodelling and mucus hypersecretion [52]. Infammation is central in COPD development and the release of infammatory mediators and destructive enzymes by infammatory cells implicated in the progressive destruction of the lung in COPD [53]. Exposure to LPS and cigarette smoke can thus harm the lung tissue while inducing immune cell-mediated inflammatory and stress responses, with apoptosis and phagocytic activity serving to help remove injured cells. In addition, all cell subtypes exhibited enrichment for the mRNA processing, formation of the ternary complex, and 43S complex signaling pathways in our study (Fig 3d). We suspect, exposure to LPS and cigarette smoke were associated with apoptotic cell death and proliferative activity, playing a dual role in regulating cell activity in the context of COPD induction.

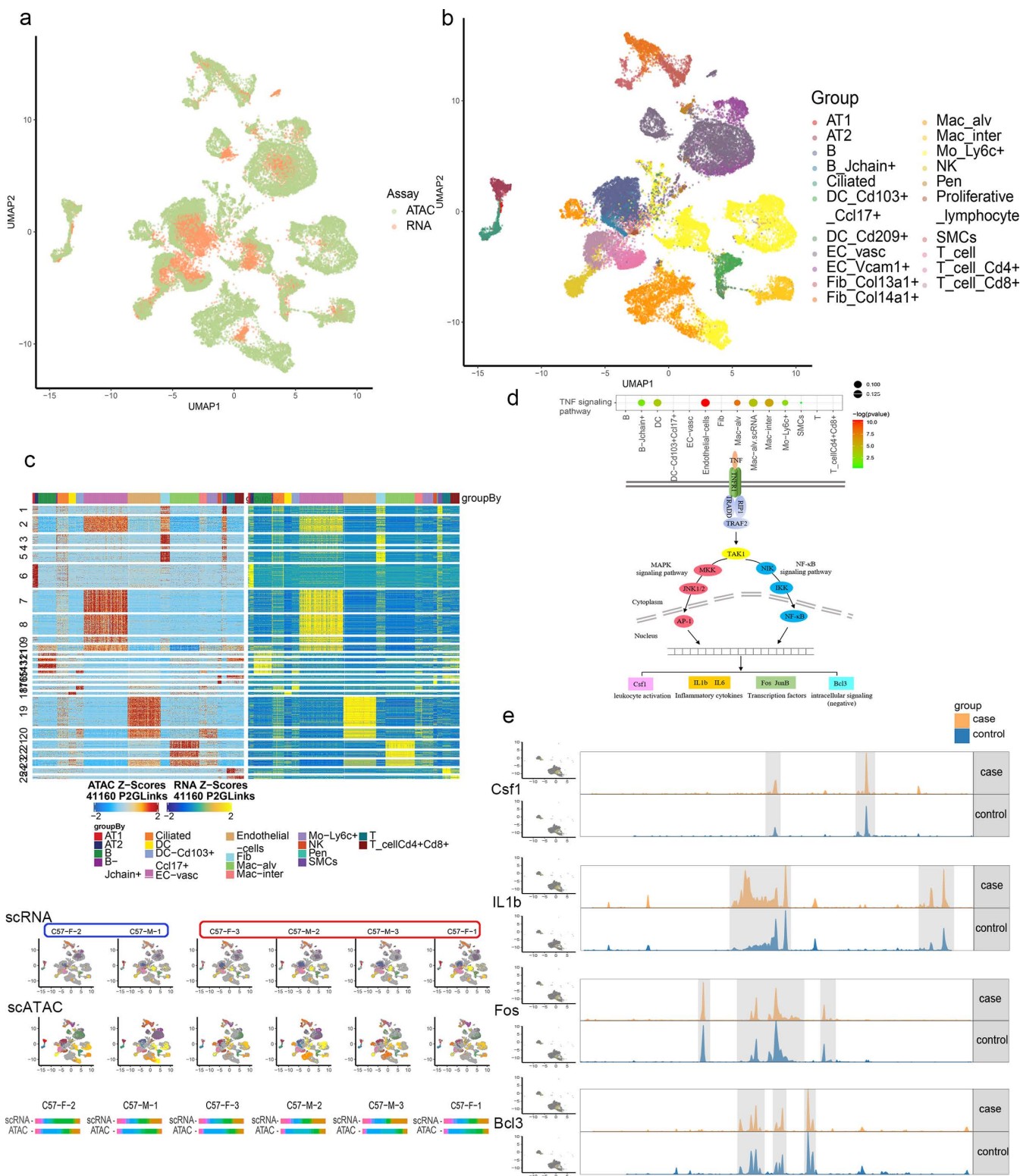

**Fig 4. Integration of scATAC-seq and scRNA-seq data in the mouse lung tissues. a: Integration UMAP plot of scRNA-seq and scATAC-seq data.** Colors indicate each dataset; b: Integration UMAP plot of scRNA-seq and scATAC-seq data. Colors represent different cell types. c: Heatmap showing the gene scores (left) and gene expression (right) of cell type-specific expression genes across 200 pseudobulk samples (Methods). Rows were

clustered using k-means clustering (k = 20). For visualization, 10,000 rows were randomly sampled (top). Cell profile of each sample for both scRNA-seq and scATAC-seq data (Middle). The proportion of cells in each sample that were broadly classified cell type (bottom); d: Cell types of TNF signaling pathway-enriched and schematic diagram of the TNF Signaling pathway; e: Specific gene expression profile of scRNA-seq data and corresponding tracks displaying the aggregate accessibility of scATAC-seq data in case and control group.

These comprehensive scATAC-seq and scRNA-seq analyses revealed a high degree of consistency regarding cell types (Fig 4b), differentially enriched genes (Fig 4c), and signaling pathways. In particular, the TNF signaling pathway was enriched in both datasets, with consistent changes in endothelial cells, mononuclear cells/macrophages, B cells, and DCs (Fig 4d). TNF acts as a key inflammatory mediator crucial for immunoregulatory activities and cytotoxic effects [54]. TNF, predominantly produced by monocytes, macrophages, and activated T cells, is activated strongly by LPS. In our study, COPD model mice were established by LPS induction and cigarette smoke exposure. COPD patients have previously been shown to exhibit elevated pulmonary TNF-α concentrations as compared to healthy individuals, and TNF-α has been linked to COPD progression [55]. Overexpression of TNF-α can lead to emphysema, pulmonary fibrosis, and muscle mass reduction in COPD patients [56]. TNF signaling also plays a key role in other lung-related diseases. TNF pathway was the differential KEGG pathway in the ovalbumin (OVA)-induced allergic asthmatic mice compared to healthy mice and the OVA-induced mice compared to the IL-17A knockout OVA-induced mice [57]. The TNF-α level was shown to increase significantly and remain steadily high during the development of acute lung injury (ALI), and that intranasal administration of an aptamer targeting TNF-α, conjugated with polyethylene glycol, to mice with ALI suppresses the development of an inflammation in the respiratory system of experimental animals [58]. The TNF signaling pathway maybe an important mechanism in the occurrence and development of COPD and other inflammatory lung diseases.

In tumorigenesis, TNFR plays important roles in multiple aspects of tumor progression, including the proliferation of cancer cells, the evasion of immune surveillance, the activation of endothelial cells and angiogenesis, and the formation of a pre-metastasis milieu [25,59]. TNFR could be an important mediatory factor in lung cancer development [60]. TNFR1 upregulation in alveolar macrophages and active cellular communication was observed in our data (Fig 4d). scATAC-seq data showing increased accessibility of *Bcl3*, *IL1b*, *Fos*, and *Csf1* loci, aligning with scRNA-seq findings of increased expression in COPD model mice compared to control mice (Fig 4e). Csf1 encoded a cytokine that plays an essential role in the regulation of survival, proliferation and differentiation of macrophages and monocytes, and promotes the release of pro-inflammatory chemokines, and plays an important role in innate immunity and in inflammatory processes [61]. IL1b is a potent pro-inflammatory cytokine, induces T-cell activation, Th17 differentiation, B-cell activation and cytokine production [62]. Bcl3, acts as transcriptional activator that promotes transcription of NF-κB target genes in the nucleus [63]. Fos proteins that can dimerize with proteins of the Jun family, thereby forming the transcription factor complex AP-1, the Fos proteins have been implicated as regulators of cell proliferation, differentiation, and apoptosis [64]. Our research suggests that, TNF signaling affects differentiation and activation of macrophages, inducing an inflammatory response that affects endothelial cells, thereby undermining the structural integrity of the lung and causing further inflammation.

There are certainly some limits of our current study, including 1) sample collection based on partial lung tissues from the mice, not the whole lung tissues of mouse, 2) limited number of subjects involved, and 3) the time point bias in the process of COPD model mice building. Although we found that the TNF signaling associated with the development of this disease in chronic obstructive pulmonary disease model mice, our findings need to be fully verified by future in vivo experiments.

In summary, via analyzing the scRNA-seq data and scATAC-seq of the lung tissues from COPD model mice, we identified mononuclear cells/macrophages, B cells, and DCs cells as the most COPD-, and smoking-associated cell types. Advanced bioinformatic analyses further revealed the TNF signaling pathway might be associated with COPD incidence. Our research makes an important supplement to the existing pathological mechanisms, and is expected to bring new strategies for the prevention and treatment of COPD.

## Supporting information

**S1 Fig. UMAP plots showing the single-cell ATAC gene-activity scores for cluster-specific markers.** Colors from dark blue to yellow represent the gene score from low to high level.
(PDF)

**S2 Fig. Heatmap showing the expression levels of DEGs between case and control group.** Colors on the top show the different cell types while the bottom color means the case(red) and control(blue) group.
(PDF)

**S3 Fig. Genome accessibility track visualization of IL10 with peak co-accessibility in case (up) and control (down) of specific cell types (B cell, B-j-chain+, DC, DC+).** Cell types are labeled on the top of each column. Colors from grey to dark represent the accessibility levels from low to high.
(PDF)

**S4 Fig. Genome accessibility track visualization of CD44 with peak co-accessibility in case (up) and control (down) of specific cell types (B j-chain+, fib, Mac-alv, T cell CD4+CD8+).** Cell types are labeled on the top of each column. Colors from grey to dark represent the accessibility levels from low to high.
(PDF)

**S5 Fig. Bubble plot showing gene expression of each cell type markers in annotated clusters.** The expression of identifying markers is sometimes evident in several clusters. For each group of markers, the dot size indicates the mean fraction of cells expressing the markers. Color indicates mean expression level.
(PDF)

**S6 Fig. Network plots showing the changes in ligand-receptor interaction events between different cell types in the case and control comparison groups.** Cell-cell communication is indicated by the connected line. The thickness of the lines is positively correlated with the number of ligand-receptor interaction events.
(PDF)

**S1 Table. Single-cell ATAC-seq QC statistics.**
(DOCX)

**S2 Table. Peak number statistics.**
(DOCX)

## Acknowledgments

The authors would like to thank all worker who participated in this study.

## Author contributions

**Conceptualization:** lindong Yuan, mingliang Gu, yan Wang, Xiaodong Jia.

**Data curation:** Xiaodong Jia.

**Formal analysis:** Li Zhou, ruihua zhang.

**Methodology:** qiang zhang, Li Zhou, ruihua zhang, shanshan Pan, xizi Wang, lili Yi, fengjiao Yuan, xianchao Guo, Xiaodong Jia.

**Software:** xianchao Guo, Xiaodong Jia.

**Validation:** lindong Yuan, yan Wang.

**Visualization:** qiang zhang, Li Zhou, ruihua zhang, xizi Wang.

**Writing – original draft:** qiang zhang, Li Zhou, ruihua zhang.

**Writing – review & editing:** qiang zhang.

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
