## [Decision Letter · Decision Letter 0]

6 Nov 2024

PONE-D-24-39675scRNA-seq and scATAC-seq analyses highlight the role of TNF signaling pathway in chronic obstructive pulmonary disease model micePLOS ONE

Dear Dr. Jia,

Thank you for submitting your manuscript to PLOS ONE. After careful consideration, we feel that it has merit but does not fully meet PLOS ONE’s publication criteria as it currently stands. Therefore, we invite you to submit a revised version of the manuscript that addresses the points raised during the review process.

We look forward to receiving your revised manuscript.

Kind regards,

Guanghui Liu

Academic Editor

PLOS ONE

Journal Requirements:

2. Thank you for stating the following financial disclosure: [This study was supported by The National Natural Science Foundation of China (Grant No. 82000066), and the Project of Medical and Health Science and Technology in Shandong Province(Grant No.202303021557) and the Traditional Chinese Medicine Science and Technology Project of Shandong Province (Grant No.Z-2023033)].

Reviewers' comments:

Reviewer's Responses to Questions

**Comments to the Author**

1. Is the manuscript technically sound, and do the data support the conclusions?

Reviewer #1: Yes

Reviewer #2: Partly

2. Has the statistical analysis been performed appropriately and rigorously? 

Reviewer #1: Yes

Reviewer #2: N/A

3. Have the authors made all data underlying the findings in their manuscript fully available?

Reviewer #1: Yes

Reviewer #2: Yes

4. Is the manuscript presented in an intelligible fashion and written in standard English?

Reviewer #1: Yes

Reviewer #2: Yes

5. Review Comments to the Author

Reviewer #1: This manuscript explores the role of the TNF signaling pathway in chronic obstructive pulmonary disease (COPD) using scRNA-seq and scATAC-seq analyses in model mice. The integration of these high-throughput techniques provides a comprehensive view of the transcriptomic and chromatin accessibility changes associated with COPD, focusing particularly on the TNF pathway's role in inflammation and immune cell function. The study is methodologically sound, and the findings are significant as they contribute to understanding the molecular mechanisms underpinning COPD, potentially guiding targeted therapies.

The dual use of scRNA-seq and scATAC-seq provides a robust dataset for analyzing gene expression and regulatory elements in COPD.Detailed methodological description enhances the reproducibility of the research.Significant insights into the role of the TNF signaling pathway in COPD pathogenesis, supported by comprehensive data analysis.

The manuscript would benefit from a more extensive review of the literature on TNF signaling in other inflammatory and pulmonary conditions to position its findings within a broader context. In the introduction, mode details should be mentioned for cancer treatment, such as referring to “Cancer treatments: Past, present, and future, 2024”. Additional comparisons with other studies that have used similar methodologies in different diseases could highlight the unique contributions of this work to the field of pulmonary research. The paper should refer to other studies using scRNA seq analysis, such as “Identification of the novel exhausted T cell CD8 + markers in breast cancer, 2024” and scATAC-seq such as “Integrative Single-Cell RNA-Seq and ATAC-Seq Analysis of Human Developmental Hematopoiesis, 2021”

The manuscript presents valuable findings that add significantly to our understanding of the molecular underpinnings of COPD, particularly through the lens of the TNF signaling pathway. However, enhancing the literature review, providing more detailed methodological justifications, and expanding on the clinical implications and limitations of the study would improve its impact and relevance. These improvements would not only solidify the current findings but also help pave the way for future research and potential clinical applications.

Reviewer #2: 1. In the abstract section, the term scATAC should not be abbreviated at first mention.

2. In the introduction L 48, the HHIP should be given completely when first mentioned.

3.From line 56 to L 66, the paragraph is mostly methodology and conclusion. The purpose of the study is not elaborated. The same goes to the importance of HHIP and scATAC in COPD study.

4. Do you think 2 mice for each group is enough for accurate experiment and data?

5. The discussion is general and mostly repeating to results. You need to signify each of the parameters studied in initiating COPD.

6. PLOS authors have the option to publish the peer review history of their article (what does this mean? ). If published, this will include your full peer review and any attached files.

**Do you want your identity to be public for this peer review?** For information about this choice, including consent withdrawal, please see our Privacy Policy .

Reviewer #1: **Yes: ** Lei Zhang

Reviewer #2: No

---

## [Author Response · Author response to Decision Letter 1]

21 Nov 2024

Reviewer #1: 

This manuscript explores the role of the TNF signaling pathway in chronic obstructive pulmonary disease (COPD) using scRNA-seq and scATAC-seq analyses in model mice. The integration of these high-throughput techniques provides a comprehensive view of the transcriptomic and chromatin accessibility changes associated with COPD, focusing particularly on the TNF pathway's role in inflammation and immune cell function. The study is methodologically sound, and the findings are significant as they contribute to understanding the molecular mechanisms underpinning COPD, potentially guiding targeted therapies. The dual use of scRNA-seq and scATAC-seq provides a robust dataset for analyzing gene expression and regulatory elements in COPD.Detailed methodological description enhances the reproducibility of the research.Significant insights into the role of the TNF signaling pathway in COPD pathogenesis, supported by comprehensive data analysis.

The manuscript would benefit from a more extensive review of the literature on TNF signaling in other inflammatory and pulmonary conditions to position its findings within a broader context. In the introduction, mode details should be mentioned for cancer treatment, such as referring to “Cancer treatments: Past, present, and future, 2024”.

Response: Thank you for your comments. As you suggest, we have made revisions in our manuscript, focus on the introduction and discussion sections. We reviewed the relevant literature, and added the relevant studies on TNF signaling pathway in diseases, such as cancer (Lung cancer, Breast cancer), allergic asthmatic, and lung injury.

Additional comparisons with other studies that have used similar methodologies in different diseases could highlight the unique contributions of this work to the field of pulmonary research. The paper should refer to other studies using scRNA seq analysis, such as “Identification of the novel exhausted T cell CD8 + markers in breast cancer, 2024” and scATAC-seq such as “Integrative Single-Cell RNA-Seq and ATAC-Seq Analysis of Human Developmental Hematopoiesis, 2021”

Response: Thank you for your comments. We have read the reference “Integrative Single-Cell RNA-Seq and ATAC-Seq Analysis of Human Developmental Hematopoiesis, 2021”. The data analysis strategy of the reference is basically the same as our data analysis strategy, only the subheadings of each method are named differently. For example, “Upstream analysis of RNA-Seq data” mentioned in the reference corresponds to our method “RNA-Seq raw data processing”; “Upstream analysis of scATAC-Seq data” corresponding to our method “scATAC-seq data pre-processing” and “scATAC-seq quality control”; “Dimensionality reduction of scATAC-Seq data” corresponding to our method “scATAC-seq batch alignment and dimensionality reduction”; “Cell type classification” corresponding to our method “scRNA-seq cell clustering and identification”; “Differential expression analysis” corresponding to our method “DEG identification and analysis”; “Integration of scRNA-Seq and scATAC-Seq data” corresponding to our method “scATAC-seq and scRNA-seq dataset integration”.

The manuscript presents valuable findings that add significantly to our understanding of the molecular underpinnings of COPD, particularly through the lens of the TNF signaling pathway. However, enhancing the literature review, providing more detailed methodological justifications, and expanding on the clinical implications and limitations of the study would improve its impact and relevance. These improvements would not only solidify the current findings but also help pave the way for future research and potential clinical applications.

Response: Thank you for your comments. As you suggest, we reviewed the relevant literature, and supplemented the relevant research on TNF signaling pathway in lung disease. Meanwhile, in the penultimate paragraph of the discussion, we discussed the limitations of this study.

Reviewer #2: 

1.In the abstract section, the term scATAC should not be abbreviated at first mention.

Response: Thank you for your comments. We have added the full name of scATAC. Please see the abstract section.

2. In the introduction L 48, the HHIP should be given completely when first mentioned.

Response: Thank you for your comments. We have added HHIP's full name. Please see line 49.

3.From line 56 to L 66, the paragraph is mostly methodology and conclusion. The purpose of the study is not elaborated. The same goes to the importance of HHIP and scATAC in COPD study.

Response: Thank you for your comments. We have added the purpose of this study. Please see line 69 to L 71.

4. Do you think 2 mice for each group is enough for accurate experiment and data?

Response: As you said, this study included a relatively small sample size. Indeed, We used eight mice (4 male and 4 female) to create animal models. It is a pity that relatively comprehensive experimental arrangements was not performed in the process of model building and sample collection due to feeding, transportation and other reasons.

5. The discussion is general and mostly repeating to results. You need to signify each of the parameters studied in initiating COPD.

Response: Thank you for your comments. As you suggest, we have made revisions in our manuscript, focus on the discussion section. Meanwhile, we have added the relevant studies on TNF signaling pathway in lung diseases.

---

## [Decision Letter · Decision Letter 1]

10 Mar 2025

PONE-D-24-39675R1scRNA-seq and scATAC-seq analyses highlight the role of TNF signaling pathway in chronic obstructive pulmonary disease model micePLOS ONE

Dear Dr. Jia,

Thank you for submitting your manuscript to PLOS ONE. After careful consideration, we feel that it has merit but does not fully meet PLOS ONE’s publication criteria as it currently stands. Therefore, we invite you to submit a revised version of the manuscript that addresses the points raised during the review process.

We look forward to receiving your revised manuscript.

Kind regards,

Tomasz W. Kaminski

Academic Editor

PLOS ONE

**Journal Requirements:**

Reviewers' comments:

Reviewer's Responses to Questions

**Comments to the Author**

1. If the authors have adequately addressed your comments raised in a previous round of review and you feel that this manuscript is now acceptable for publication, you may indicate that here to bypass the “Comments to the Author” section, enter your conflict of interest statement in the “Confidential to Editor” section, and submit your "Accept" recommendation.

Reviewer #2: All comments have been addressed

Reviewer #3: (No Response)

2. Is the manuscript technically sound, and do the data support the conclusions?

Reviewer #2: Yes

Reviewer #3: Yes

3. Has the statistical analysis been performed appropriately and rigorously? 

Reviewer #2: Yes

Reviewer #3: Yes

4. Have the authors made all data underlying the findings in their manuscript fully available?

Reviewer #2: Yes

Reviewer #3: Yes

5. Is the manuscript presented in an intelligible fashion and written in standard English?

Reviewer #2: Yes

Reviewer #3: Yes

6. Review Comments to the Author

**Reviewer #2:**  (No Response)

**Reviewer #3: ** In this work, the authors establish and then generate from a mouse COPD model single cell ATAC-seq and RNA-seq datasets. They interrogate these data using an array of current computational biology approaches. They find that immune response cell such as B cells, mononuclear cells/ macrophages and dendritic cells are associated with COPD. They also find that TNF signaling may play a key role in the disease.

The work done, beginning with creation of the model, is thorough and the characterizations of the model are, potentially, a worthy contribution. The manuscript is generally well written. Moreover, the authors have been careful to point out the limitations of the study. The major difficulty I have with this work is that, beyond the novel combination of approaches, the authors do not highlight the gap in knowledge that this study’s findings help fill (if any). For example, there is previous evidence of an association of TNF signaling with COPD in the literature, so there is a need to highlight what is novel here.

Some suggested edits:

-pages need to be numbered

-on line 198, regarding the use of the Seurat package’s FindAllMarkers function, it is not specified what test was used (for example Wilcox, t-test, DESeq2, or other)

-on line 200, enrichKEGG from clusterProfiler could not have been used for Gene Ontology enrichment

-on line 210, please specify the source of the addGeneIntegrationMatrix function i.e. ArchR

-on line 265 “based” should replace “base”

-on line 286 “There are color represents” needs to be corrected

-lines 309 -312: please specify the figure numbers that are relevant here (was it Figure 3d?). Indeed in much of the Discussion section, refernces to results are made without stating what the associated Figure numbers are

-lines 323 et seq: Mouse gene symbols should be “Sox7, Hoxa5”, etc… rather than the human “SOX7, HOXA5”, etc.. as was stated. Moreover, gene symbols need to be italicized

-lines 409 – 417: A number of major statements are made without the requisite literature citations

7. PLOS authors have the option to publish the peer review history of their article (what does this mean? ). If published, this will include your full peer review and any attached files.

**Do you want your identity to be public for this peer review?** For information about this choice, including consent withdrawal, please see our Privacy Policy .

Reviewer #2: **Yes: ** Rebah N. Algafari

Reviewer #3: No

---

## [Author Response · Author response to Decision Letter 2]

20 Mar 2025

Reviewer #3:

In this work, the authors establish and then generate from a mouse COPD model single cell ATAC-seq and RNA-seq datasets. They interrogate these data using an array of current computational biology approaches. They find that immune response cell such as B cells, mononuclear cells/ macrophages and dendritic cells are associated with COPD. They also find that TNF signaling may play a key role in the disease.

The work done, beginning with creation of the model, is thorough and the characterizations of the model are, potentially, a worthy contribution. The manuscript is generally well written. Moreover, the authors have been careful to point out the limitations of the study. The major difficulty I have with this work is that, beyond the novel combination of approaches, the authors do not highlight the gap in knowledge that this study’s findings help fill (if any). For example, there is previous evidence of an association of TNF signaling with COPD in the literature, so there is a need to highlight what is novel here.

Response: Thank you for your comment. As you said, we do not highlight the gap in knowledge that our study’s findings help fill. Indeed, animal models and research methods are the innovations of our study. Our study provides the data of scRNA-seq and scATAC-seq for COPD model mice. Through the joint analysis of scRNA-seq and scATAC-seq, the cell map was systematically mapped, and from the aspects of differential genes, signaling pathways and cell communication to understand the pathogenesis of COPD. TNF signaling has been reported to be associated with COPD. Our study also found that TNF signaling, which is consistent with previous studies. In the last paragraph of the introduction, the first and last paragraph of the discussion section, we describe the key points and key results of this study.

Some suggested edits:

-pages need to be numbered

Response: Thank you for your comments. We paginated our manuscript.

-on line 198, regarding the use of the Seurat package’s FindAllMarkers function, it is not specified what test was used (for example Wilcox, t-test, DESeq2, or other)

Response: Thank you for your comment. In our analysis using the FindAllMarkers function from the Seurat package, we used the default statistical test, which is the Wilcoxon rank-sum test (as documented in Seurat). This method identifies differentially expressed genes between clusters. We have made revisions in our manuscript.

-on line 200, enrichKEGG from clusterProfiler could not have been used for Gene Ontology enrichment

Response: Thank you for your comment. You are absolutely correct that the enrichKEGG function from the clusterProfiler package is specifically designed for KEGG pathway enrichment analysis and cannot be used for Gene Ontology (GO) enrichment. For our GO enrichment analysis, we actually used the enrichGO function from clusterProfiler. We acknowledge that this was not clearly described in the original manuscript, which may have caused confusion. As you suggest, we have made revisions in our manuscript.

-on line 210, please specify the source of the addGeneIntegrationMatrix function i.e. ArchR

Response: Thank you for your suggestion. We confirm that the addGeneIntegrationMatrix function is from the ArchR package. We agree that explicitly stating the source would enhance the clarity of the manuscript. Therefore, we have updated the text.

-on line 265 “based” should replace “base”

Response: Thank you for your comments. We have made revisions in our manuscript.

-on line 286 “There are color represents” needs to be corrected

Response: Thank you for your comments. We have made revisions in our manuscript.

-lines 309 -312: please specify the figure numbers that are relevant here (was it Figure 3d?). Indeed in much of the Discussion section, refernces to results are made without stating what the associated Figure numbers are

Response: Thank you for your comments. As you suggest, we have made revisions in our manuscript.

-lines 323 et seq: Mouse gene symbols should be “Sox7, Hoxa5”, etc… rather than the human “SOX7, HOXA5”, etc.. as was stated. Moreover, gene symbols need to be italicized

Response: Thank you for your comments. We have made revisions in our manuscript (line 323). Meanwhile, the gene symbols that appear in our manuscript are shown in italics.

-lines 409 – 417: A number of major statements are made without the requisite literature citations

Response: Thank you for your comments. As you suggest, We have added the relevant references.

---

## [Editor Report · Decision Letter 2]

25 Mar 2025

scRNA-seq and scATAC-seq analyses highlight the role of TNF signaling pathway in chronic obstructive pulmonary disease model mice

PONE-D-24-39675R2

Dear Dr. Jia,

We’re pleased to inform you that your manuscript has been judged scientifically suitable for publication and will be formally accepted for publication once it meets all outstanding technical requirements.

Kind regards,

Tomasz W. Kaminski

Academic Editor

PLOS ONE

Additional Editor Comments (optional):

Thank you for submitting the revised version of your manuscript. After reviewing the changes, I confirm that all the minor issues raised in the previous round have been addressed. The manuscript is now suitable for acceptance.

Please note that there are still some minor typos that will need to be corrected during the proofreading stage. Otherwise, no further revisions are required.

Best Regards.

---

## [Editor Report · Acceptance letter]

PONE-D-24-39675R2

PLOS ONE

Dear Dr. Jia,

I'm pleased to inform you that your manuscript has been deemed suitable for publication in PLOS ONE. Congratulations! Your manuscript is now being handed over to our production team.

Kind regards,

on behalf of

Dr. Tomasz W. Kaminski

Academic Editor

PLOS ONE